# Lower limb amputations among individuals living with diabetes mellitus in low- and middle-income countries: A systematic review protocol

Eyitayo Omolara Owolabi[1], Davies Adeloye[2], Anthony Idowu Ajayi [3]*, Michael McCaul[4], Justine Davies[5], Kathryn M. Chu[1]

1 Centre for Global Surgery, Department of Global Health, Stellenbosch University, Stellenbosch, Cape Town, South Africa, 2 Centre for Global Health, Usher Institute, University of Edinburgh, Edinburgh, United Kingdom, 3 Population Dynamics and Sexual and Reproductive Health Unit, Africa Population and Health Research Center, Nairobi, Kenya, 4 Division of Epidemiology and Biostatistics, Department of Global Health, Stellenbosch University, Stellenbosch, Cape Town, South Africa, 5 Institute of Applied Health Research, University of Birmingham, Birmingham, United Kingdom

* ajayianthony@gmail.com

**Funding:** The author(s) received no specific funding for this work.

## Abstract

### Background

The burden of diabetes mellitus (DM) and its associated complications continue to burgeon, particularly in low- and middle-income countries (LMICs). Lower limb amputation (LLA) is one of the most life-altering complications of DM, associated with significant morbidity, mortality and socio-economic impacts. High-income countries have reported a decreasing incidence of DM-associated LLA, but the situation in many LMICs is unknown. We aim to conduct a systematic review to determine the incidence and prevalence of DM-associated LLA in LMICs to better inform appropriate interventions and health system response.

### Methods and analysis

A systematic search of the literature will be conducted on five databases: MEDLINE, Embase, Cumulative Index of Nursing and Allied Health Literature (CINAHL), Scopus and African Journal Online (AJOL). Only observational, quantitative studies reporting the incidence and/or prevalence of DM-related LLA will be considered. A validated study design-specific critical appraisal tool will be used to assess the risk of bias in individual studies. We will determine the incidence of LLA by examining the number of new cases of LLA among individuals with confirmed DM diagnosis during the specified period, while the prevalence will be based on the total number of all new and existing LLAs in a population. LLA will be considered as the resection of the lower limb from just above the knee to any point down to the toe. If heterogeneity is low to moderate, a random-effects meta-analysis will be conducted on extracted crude prevalence/incidence rates, with the median and interquartile range also reported. The systematic review will be performed in accordance with the JBI

**Competing interests:** The authors have declared that no competing interests exist.

guideline for prevalence and incidence review. Study reporting will follow the Preferred Reporting Items for Systematic Review and Meta-Analyses (PRISMA) guideline.

## Prospero registration number

CRD42021238656.

## Introduction

The burden of Diabetes Mellitus (DM) continues to grow, especially in low- and middle-income countries (LMICs) [1]. Since 1980, the number of persons living with DM globally has nearly quadrupled [1], with almost 80% of them residing in LMICs [2]. The rising DM prevalence is largely attributable to demographic transition, population ageing, obesity and unhealthy lifestyle behaviours [1].

DM leads to several systemic complications, with lower limb amputation (LLA) being one of the most life altering [3]. Specifically, DM contributes to the development of microvascular and macrovascular complications such as peripheral arterial disease (PAD) and peripheral neuropathy (PN) which predispose to non-healing, infected or gangrenous foot ulcers, which ultimately lead to LLA [4, 5]. The risk of PAD, PN and LLA significantly increases with longer duration and poorly controlled DM [4]. LLA risk among individuals with poorly controlled DM is 26% higher with every percentage increase in glycated haemoglobin (HbA1c), a measure of glycaemic status [6].

LLA poses a substantial health, psychological [7] and socio-economic burden, both at individual and country level [8–10], and greatly affects quality of life [11]. It is also associated with disability and premature mortality [12]. In 2016, there were an estimated 1.5 million years lived with disability (YLD) resulting from amputations, a 32% increase from 1990 [13]. Not only does LLA have deleterious impacts in its own right, it is also associated with increased risk for other DM complications such as cardiovascular diseases [14].

The burden of diabetes-related complications is especially high across LMICs [1], where access to quality DM care is limited [15]. With regard to LLA among persons living with diabetes, perhaps, one of the challenges in the response so far has been a poor understanding and awareness of the disease epidemiology in LMICs, which limits targeted response. While individual studies on LLA prevalence and incidence do exist in LMICs [16–19], no study has synthesised data from this region. Existing reviews had a global inclusion criteria [20, 21] but only one of them included few LMICs in their final analysis but did not follow a systematic approach [21]. Reasons for low or no representation of LMICs in previous studies is not known and estimates from these studies may not accurately reflect the situation in LMICs. In this study, we will define LMICs according to the World Bank income categories [22], taking care to apply our data synthesis and analyses to the WHO regions. This, to the best of our knowledge, will be the first systematic review of LLA among persons with diabetes in LMICs using this approach. This study therefore offers up-to-date data on the incidence and prevalence of diabetes-related LLA that can inform a much-needed response, priorities setting and further research.

### Aim and objectives

We aim to conduct a systematic review to determine the incidence and prevalence of DM-related LLA in LMICs.

Secondary objectives of the systematic review include:

1. to determine the incidence of re-amputations and contralateral amputations and, where possible;

2. to investigate time trends in the incidence and prevalence of LLA among DM individuals in LMICs.

## Methods

This systematic review on the incidence and prevalence of LLA in LMICs will follow the Joanna Briggs Institute (JBI) methodology for systematic reviews of prevalence and incidence [23]. The study protocol is in accordance with the PRISMA protocol checklist for systematic reviews [24, 25].

### Eligibility criteria

**Inclusion criteria.**   *Types of studies*. Prospective and retrospective observational studies, specifically, descriptive and analytic cross- sectional studies reporting the incidence and/or prevalence of lower limb amputations among individuals with confirmed DM diagnosis for a specified period will be included. Also included will be randomised controlled trials (RCTs) with baseline data providing prevalence estimates. Studies involving both DM and non-DM individuals will also be included as long as estimates of LLA among the DM population are provided. There is no age or gender restriction for the participants. For duplicate or overlapping studies, the study with the most detailed data will be included, with additional data, if necessary, extracted from others.

*Context*. This review will be limited to studies conducted in LMICs. Countries currently listed by the World Bank as low-income, lower-middle-income and upper-middle-income based on income levels, will be considered LMICs [22].

*Language*. Only studies written in English or with an available English version, or with the possibility of translating the full article via Google Translate, will be included.

*Study period*. Our focus will be on the past three decades; we will therefore include studies conducted from 1990 until 28[th] February 2022.

**Exclusion criteria.**   • Qualitative studies on LLA among persons with diabetes;

• Studies that reported traumatic LLA or cancer-related LLA;

• Case reports, case series, case-control, reviews, viewpoints, letters or opinion-based articles, and

• RCTs without a representative population denominator.

### Information sources and search strategy

**Electronic searches.**   A comprehensive systematic search of the literature will be conducted on five databases: MEDLINE, Embase, Scopus, CINAHL, and African Journals Online (AJOL) using a combination of MeSH terms and keywords such as "diabetes", "amputation", "lower limb amputation", "lower extremity amputation", "amputee" and "incidence" (see S1 Appendix). Terms representing similar concepts will be separated using the Boolean operator "OR" while terms representing different concepts will be separated using the Boolean operator

"AND". Hand searching for relevant articles from the reference list of reviews and included studies will be conducted. Where the full text of relevant studies cannot be retrieved, efforts will be made to request the full text from study corresponding authors.

**Searching of other sources.** Unpublished studies/grey literature will be hand searched through Google Scholar and grey literature databases such as WHO Library, OpenSIGLE and Open Grey using the study keywords.

## Study records

### Data management and study selection process

Articles retrieved from databases will be imported in Endnote to remove duplicates. Articles will then be exported into Covidence Systematic Review software (Veritas Health Innovation, Melbourne, Australia) for data management.

After duplicates have been removed from search results, initial screening of titles and abstracts will be conducted independently by two authors following the selection criteria; discrepancies will be resolved by discussion or with a third reviewer. Full text-review of the included studies will also be conducted independently by two authors. All disagreements will be resolved by discussion between two authors and if required, with a third reviewer. The reasons for the exclusion of full text articles will be given. A detailed description and summary of the search results and the selection process will be presented using a PRISMA flow diagram. Details of all the included studies will be reported in a table of included studies.

### Data extraction

The standardised JBI data extraction tool for prevalence and incidence review will be adapted [23]. The data extraction tool will be pilot tested using three randomly selected (included) full-text articles after which the tool will be refined and adjusted as necessary. Double (independent) extraction of data will be conducted, and the extracted information will be compared. Where a discrepancy exists between the two extractions, both authors will re-assess identified studies and correct any errors. When necessary, study authors will be contacted to request missing information for clarifications.

### Data items and description

**General study information.** General study information to be extracted include the first and last authors' names, study title, year of publication, the main affiliation of the primary and senior authors and the data source.

**Study characteristics.** Study characteristics to be extracted include study population (described by age and sex), study setting/country, study design and period, funding source, year of data collection, sample size, and study denominator (population-level or health facility).

**Outcome data.** We will extract data on LLA type and location, number of DM individuals in the setting at the specified time, the incidence of LLA reported among DM individuals, the prevalence of LLA reported among DM individuals, the type of LLA (major, minor or both), with a description of it (first, re-amputation or contralateral amputation), the confidence interval and the absolute number of cases.

### Case definitions

DM will be defined as specified by the study authors; we will capture the criteria used by authors in order to appraise heterogeneity.

LLA will be defined as the resection of a segment of the lower limb through the tibia and fibula, knee or ankle joint, or femur down to the toe. Major amputations are resections proximal to the tarsometatarsal joint while minor amputations are resections occurring distal or through the tarsometatarsal joint [26].

## Study outcomes and prioritisation

The primary outcome of this review is the incidence and prevalence of DM-related LLA in LMICs. Incidence will be defined as the number of new cases of LLA divided by the total DM population in the study during the specified period. Incidence measures will include incidence rate and cumulative incidence. Reporting of incidence at person level will be one amputation per person (the first or the highest) and both levels of amputation, major and minor will be considered. Prevalence will be defined as the total number of DM-related LLAs in the population for a given period.

## Assessment of methodological quality of studies

Eligible studies reporting prevalence data that meet the inclusion criteria will be critically appraised for methodological quality using the validated JBI critical appraisal tool for prevalence studies [23]. Studies reporting incidence data will be appraised using a validated tool appropriate for the study design. For instance, cohort studies will be critically appraised using the validated Newcastle-Ottawa quality appraisal tool for non-randomised studies [27]. Critical appraisals will be done at the study level by two independent reviewers. Discrepancies will be resolved by discussion, or with a third reviewer. Studies rated as high (score of >8 of 10) to moderate quality (score of 6–8) will be included in the analysis while low-quality studies (score of 0–5) will be excluded.

## Data analysis and presentation of results

The median and interquartile range of all crude incidence rates will be reported. A Freeman–Tukey double-arcsine transformation will be applied to normalise data, given an expected wide variation in extracted crude estimates from studies. Transformed data will be used to calculate summary proportion and 95% CI using a random effects model. A meta-analysis of the estimates of prevalence and incidence will be conducted if studies are sufficiently homogenous. We will assess heterogeneity using the $X^2$ and $I^2$ tests. $I^2$ cut-off values of 0%, 25%, 50% and 75% will represent no, low, moderate and high level of heterogeneity respectively [28]. We will conduct the meta-analysis following the DerSimonian and Laird method [29]. Due to potential of high level of heterogeneity, the DerSimonian-Laird random-effects model is preferred. The DerSimonian and Laird method is widely adopted in the literature and performs robustly in various scenarios including skew-normal and extreme distributions for the effects [30].

Where meta-analysis is not possible, a narrative synthesis of results will be done, using tables and figures as appropriate for data visualisation. Also, in case of heterogeneity, we will carry out sub-group analysis and meta-regression analyses to show the potential sources of heterogeneity. Sub-group analysis will show the distribution across age group, sex and study regions. There should be at least ten studies for each variable in the model for meta-regression analysis to be performed [31]. A sensitivity analysis will be performed to assess the robustness of the study estimates, enabling us to assess the effect of omitting onestudy at a time on the pooled estimate [31]. Lastly, a funnel plot asymmetry test and Egger's test will be used to assess publication bias, if there are more than ten included studies [31].

### Assessment of certainty of evidence

We will use the grading of recommendations, assessment, development and evaluation (GRADE) instrument and Gradepro Software to assess the quality of the synthesised findings. Included studies will be assessed by study design, heterogeneity, consistency, directness, precision and publication bias. The evidence will be described as high, moderate, low or very low quality.

## Discussion

The burden of DM has been projected to grow exponentially in LMICs in the near future, while the quality of care remains poor [15]. The incidence of LLA among diabetics is considered one of the markers of diabetes care quality, while the prevalence helps us to understand the extent and depth of the disease burden. Also, the worldwide prevalence of LLA is varies. To the best of our knowledge, this proposed study will be the first to provide an estimate of the incidence and prevalence of LLA among persons living with DM in LMICs. An understanding of the LLA burden will inform targeted response. Also, its distribution across various socio-demographic groups, as defined by age or gender, for instance, may help in identifying populations at risk and designing appropriate interventions targeting those groups. Likewise, variance across countries may inform future research studies which might look at the DM care in settings with a lower incidence of LLA and identify lessons that may be translated to other resource-limited settings. This review will follow a systematic approach using a predefined protocol and following standard recommendations for data extraction [32]. Potential limitations that may be encountered include heterogeneity in the individual studies related to the quality of the study designs, definitions used, or methods of assessing outcomes. We will, however, provide a detailed narrative report highlighting these discrepancies and provide suggestions for improving future studies.

## Supporting information

**S1 Appendix. Search strategy.**
(DOCX)

**S1 Checklist.**
(DOCX)

## Author Contributions

**Conceptualization:** Eyitayo Omolara Owolabi, Kathryn M. Chu.

**Methodology:** Eyitayo Omolara Owolabi, Michael McCaul.

**Project administration:** Eyitayo Omolara Owolabi.

**Supervision:** Kathryn M. Chu.

**Validation:** Kathryn M. Chu.

**Writing – original draft:** Eyitayo Omolara Owolabi, Davies Adeloye, Anthony Idowu Ajayi, Justine Davies, Kathryn M. Chu.

**Writing – review & editing:** Eyitayo Omolara Owolabi, Davies Adeloye, Anthony Idowu Ajayi, Michael McCaul, Justine Davies, Kathryn M. Chu.

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
