## [Decision Letter · Decision Letter 0]

27 Jan 2022

PONE-D-21-29205Lower limb amputations among individuals living with diabetes mellitus in low- and middle-income countries: a systematic review protocolPLOS ONE

Dear Dr. Ajayi,

Thank you for submitting your manuscript to PLOS ONE. After careful consideration, we feel that it has merit but does not fully meet PLOS ONE’s publication criteria as it currently stands. Therefore, we invite you to submit a revised major version of the manuscript that addresses the points raised during the review process. Please find these below. Please submit your revised manuscript by Mar 10 2022 11:59PM. If you will need more time than this to complete your revisions, please reply to this message or contact the journal office at plosone@plos.org. Please include the following items when submitting your revised manuscript:A rebuttal letter that responds to each point raised by the academic editor and reviewer(s). You should upload this letter as a separate file labeled 'Response to Reviewers'.A marked-up copy of your manuscript that highlights changes made to the original version. You should upload this as a separate file labeled 'Revised Manuscript with Track Changes'.An unmarked version of your revised paper without tracked changes. You should upload this as a separate file labeled 'Manuscript'.

We look forward to receiving your revised manuscript.

Kind regards,

Daoud Al-Badriyeh

Academic Editor

PLOS ONE

Journal Requirements:

Additional Editor Comments:

Please provide detailed responses to the reviewer's comments and suggestions, including clarifying the rationale behind this study in LMICs when similar studies in other settings exist already. Why/how would LLAs differ in LMICs, and what are the anticipated implications of this, needs elaboration. This is an important aspect of a study protocol.

Reviewers' comments:

Reviewer's Responses to Questions

**Comments to the Author**

1. Does the manuscript provide a valid rationale for the proposed study, with clearly identified and justified research questions?

Reviewer #1: Partly

2. Is the protocol technically sound and planned in a manner that will lead to a meaningful outcome and allow testing the stated hypotheses?

Reviewer #1: Partly

3. Is the methodology feasible and described in sufficient detail to allow the work to be replicable?

Reviewer #1: Yes

4. Have the authors described where all data underlying the findings will be made available when the study is complete?

Reviewer #1: Yes

5. Is the manuscript presented in an intelligible fashion and written in standard English?

Reviewer #1: No

6. Review Comments to the Author

You may also provide optional suggestions and comments to authors that they might find helpful in planning their study.

Reviewer #1: The study aimed to assess the incidence and prevalence of LLA as a complication of DM in LMICs. The study objectives are methodology clear, yet the significance, novelty, and implications were not highlighted clearly. There are several ideas that were repeated in different sections which negatively affected the flow and readability of the text. While the study appears to be sound, there are several grammatical errors. I advise the authors to work with a writing coach or copy editor.

Introduction: The flow of the introduction is a bit confusing and not easy to follow. There are several ideas that were repeated in this section.

Line 55-57: The sentence is long with 3 different ideas; I suggest splitting it into sentences.

Line 74: This idea was already stated earlier in the previous paragraphs.

Line 82: The sentence is long with 3 different ideas; I suggest splitting it into sentences.

Line 83: This sentence is about the decrease of LLA in HICs which was already mentioned before in the same paragraph (lines 75-77). Also, why Africa is mentioned as part of the sentence?

The last paragraph of the introduction does not clearly emphasize rationale of this study or the current research gaps.

Methods and analysis: There are several ideas that were repeated in this section.

I suggest using the Preferred Reporting Items for Systematic Reviews and Meta-analyses Protocols (PRISMA-P) guidelines for this protocol.

Line 124: Why would you limit the search to studies conducted from 1999? There is no clear epidemiological reason to limit the search to a specific time period.

In the inclusion criteria: Is there is any minimum sample size for each study or measure to be included?

In the exclusion criteria, will you exclude RCTs (which may report a prevalence measure at the baseline), case reports, or case series?

In the exclusion criteria, will you consider duplicate or overlapping studies (i.e., if two published studies are using the same population sample but there are conducting different analyses)?

Line 141: Why only from the first ten pages?

The data management section implies the same information as the screening process. Thus, I suggest merging them into one paragraph to avoid repetition.

Line 161: I suggest specifying what are study population measures that will be extracted, like "study population and its characteristics (sex, age, etc)".

Line 164: "time trend reported where available". This is not clear. What is the time trend you are going to extract is it the time since diagnosis of DM, time since LLA, years of data collection?

For the quality assessment tool, since it was adapted, a validation method should be considered before applying it (e.g., face validity, content validity, etc.).

Study outcomes paragraph: A clear definition of the outcomes should be mentioned. Also, it seems that it is written in separate lines, not as one paragraph.

Data analysis and result presentation: No clear or detailed plan for data synthesis and analysis is presented.

Line 178: I suggest removing this sentence as it is already mentioned under "data items and description".

Lines 181-183: The two sentences should be moved to the "study outcomes paragraph" as they represent a definition of the study's primary outcome (i.e., incidence).

Lines 182-183: How you are going to combine two incidence measures in the data synthesis and analysis while they have different denominators (i.e., either DM population at a country level or DM population presenting at healthcare facility). More clarification should be provided.

Lines 185-187: I think it would be good to add any potential reasons for not conducing a meta-analysis. A justification for using DerSimonian-Laird random-effects model should be stated. Also, will you consider any method for data transformation?

Discussion: This section does not reflect the significance and novelty of the current study, as well as the implications of the future findings.

Minor issues

Line 66: HbA1c was abbreviated without prior definition.

Line 59: Better to write it as "access to healthcare services"

Lines 130-133: Be consistent in the letter case of keywords.

7. PLOS authors have the option to publish the peer review history of their article (what does this mean?). If published, this will include your full peer review and any attached files.

Reviewer #1: No

---

## [Author Response · Author response to Decision Letter 0]

21 Mar 2022

Editor’s Comment: Please provide detailed responses to the reviewer's comments and suggestions, including clarifying the rationale behind this study in LMICs when similar studies in other settings exist already. Why/how would LLAs differ in LMICs, and what are the anticipated implications of this, needs elaboration. This is an important aspect of a study protocol.

Response: The burden of diabetes-related complications is especially high across LMICs where access to quality DM care is limited. With regard to LLA among persons living with diabetes, perhaps, one of the challenges in the response so far has been a poor understanding and awareness of the disease epidemiology in LMICs, which limits targeted response. While individual studies on LLA prevalence and incidence do exist in LMICs [16-19], no study has synthesised data from this region. Existing reviews had a global inclusion criteria [20, 21] but only one of them included few LMICs in their final analysis and that specific one did not follow a systematic approach [21]. Reasons for low or no representation of LMICs in previous studies is not known and estimates from these studies may not accurately reflect the situation in LMICs. In this study, we will define LMICs according to the World Bank income categories [22], taking care to apply our data synthesis and analyses to the sub-regions. This, to the best of our knowledge, will be the first systematic review of LLA among persons with diabetes in LMICs using this approach. This study therefore offers up-to-date data on the incidence and prevalence of diabetes-related LLA that can inform a much-needed response, priorities setting and further research. 

Reviewer 1

Comment 1: Introduction: The flow of the introduction is a bit confusing and not easy to follow. There are several ideas that were repeated in this section.

Line 55-57: The sentence is long with 3 different ideas; I suggest splitting it into sentences.

Response: Thank you for the correction. We have now done a professional language editing to ensure clarity and flow of the write-up. We have separated these sentences and now reads:

“The burden of Diabetes Mellitus (DM) continues to grow, especially in low- and middle-income countries (LMICs) [1]. Since 1980, the number of persons living with DM globally has nearly quadrupled [1], with almost 80% of them residing in LMICs [2]. The rising DM prevalence is largely attributable to demographic transition, population ageing, obesity and unhealthy lifestyle behaviours”

Comment 2: Line 74: This idea was already stated earlier in the previous paragraphs.

Response: We have deleted the repeated statement on line 74.

Comment 3: Line 82: The sentence is long with 3 different ideas; I suggest splitting it into sentences.

Response: We have revised this section of the manuscript, which is the study justification section and it now reads: 

“ The burden of diabetes-related complications is especially high across LMICs [1], where access to quality DM care is limited [15]. With regard to LLA among persons living with diabetes, perhaps, one of the challenges in the response so far has been a poor understanding and awareness of the disease epidemiology in LMICs, which limits targeted response. While individual studies on LLA prevalence and incidence do exist in LMICs [16-19], no study has synthesised data from this region. Existing reviews had a global inclusion criteria [20, 21] but only one of them included few LMICs in their final analysis but did not follow a systematic approach [21]. Reasons for low or no representation of LMICs in previous studies is not known and estimates from these studies may not accurately reflect the situation in LMICs. In this study, we will define LMICs according to the World Bank income categories [22], taking care to apply our data synthesis and analyses to the sub-regions. This, to the best of our knowledge, will be the first systematic review of LLA among persons with diabetes in LMICs using this approach. This study therefore offers up-to-date data on the incidence and prevalence of diabetes-related LLA that can inform a much-needed response, priorities setting and further research.”

Comment 4: Line 83: This sentence is about the decrease of LLA in HICs which was already mentioned before in the same paragraph (lines 75-77). Also, why Africa is mentioned as part of the sentence?

Response: We have revised this section as shown in response to comment 3.

Comment 5: The last paragraph of the introduction does not clearly emphasize the rationale of this study or the current research gaps.

Response: The last paragraph, that is, the justification for the study now reads:

“The burden of diabetes-related complications is especially high across LMICs [1], where access to quality DM care is limited [15]. With regard to LLA among persons living with diabetes, perhaps, one of the challenges in the response so far has been a poor understanding and awareness of the disease epidemiology in LMICs, which limits targeted response. While individual studies on LLA prevalence and incidence do exist in LMICs [16-19], no study has synthesised data from this region. Existing reviews had a global inclusion criteria [20, 21] but only one of them included few LMICs in their final analysis but did not follow a systematic approach [21]. Reasons for low or no representation of LMICs in previous studies is not known and estimates from these studies may not accurately reflect the situation in LMICs. In this study, we will define LMICs according to the World Bank income categories [22], taking care to apply our data synthesis and analyses to the sub-regions. This, to the best of our knowledge, will be the first systematic review of LLA among persons with diabetes in LMICs using this approach. This study therefore offers up-to-date data on the incidence and prevalence of diabetes-related LLA that can inform a much-needed response, priorities setting and further research.”

Methods and analysis: 

Comment 6: There are several ideas that were repeated in this section.

I suggest using the Preferred Reporting Items for Systematic Reviews and Meta-analyses Protocols (PRISMA-P) guidelines for this protocol.

Response: The section has been re-ordered using the PRISMA-P guideline.

Comment 7: Line 124: Why would you limit the search to studies conducted from 1999? There is no clear epidemiological reason to limit the search to a specific time period.

Response: Given that we are also interested in exploring trends in epidemiology, with our focus on the past three decades, we will limit our analyses to studies conducted from 1990-2022. 

Comment 8: In the inclusion criteria: Is there any minimum sample size for each study or measure to be included? 

Response: As our study will involve multiple hierarchical datasets from each study (ie., prevalence across age- and sex- specific groups in each study), we are considering population denominators across each data point, and not necessarily sample size from the study, which is set at a minimum of 50. 

Comment 9: In the exclusion criteria, will you exclude RCTs (which may report a prevalence measure at the baseline), case reports, or case series?

Response: Case reports and case series will be excluded. RCTs without a representative population denominator will be excluded.

Comment 10: In the exclusion criteria, will you consider duplicate or overlapping studies (i.e., if two published studies are using the same population sample but there are conducting different analyses)?

Response: For duplicate or overlapping studies, the study with the most detailed data will be included, and additional data, if necessary, extracted from others. We have now added a statement on this.

Comment 11: Line 141: Why only from the first ten pages?

Response: We have now removed this restriction.

Comment 12: The data management section implies the same information as the screening process. Thus, I suggest merging them into one paragraph to avoid repetition.

Response: We have now merged these sections:

“Articles retrieved from databases will be imported in Endnote to remove duplicates. Articles will then be exported into Covidence Systematic Review software (Veritas Health Innovation, Melbourne, Australia) for data management”.

Comment 13: Line 161: I suggest specifying what are study population measures that will be extracted, like "study population and its characteristics (sex, age, etc)".

Response: Study characteristics to be extracted include study population (described by age and sex), study setting/country, study design and period, funding source, year of data collection, sample size, and study denominator (population-level or health facility). We have now stated this. 

Comment 14: Line 164: "time trend reported where available". This is not clear. What is the time trend you are going to extract is it the time since diagnosis of DM, time since LLA, years of data collection?

Response: We intended to measure differences in LLA incidence over years (1990-2022), where possible. Year(s) of data collection data will be helpful to ascertain that, and we have changed to “year(s) of data collection” for clarity. 

Comment 15: For the quality assessment tool, since it was adapted, a validation method should be considered before applying it (e.g., face validity, content validity, etc.).

Response: We will now be using the validated Joanna Briggs Institute (JBI) critical appraisal tool for prevalence studies. Studies reporting incidence data will be appraised using validated tools appropriate for the study design. For instance, cohort studies will be critically appraised using the validated Newcastle-Ottawa quality appraisal tool for non-randomised studies [28]. 

Comment 16: Study outcomes paragraph: A clear definition of the outcomes should be mentioned. Also, it seems that it is written in separate lines, not as one paragraph.

Response: We have now merged the sentences as a paragraph and the definition of outcome is now provided:

“The primary outcome of this review is the incidence and prevalence of DM-related LLA in LMICs. Incidence will be defined as the number of new cases of LLA divided by the total DM population in the study during the specified period. Incidence measures will include incidence rate and cumulative incidence. Reporting of incidence at person level will be one amputation per person (the first or the highest) and both levels of amputation, major and minor will be considered. Prevalence will be defined as the total number of DM-related LLAs in the population for a given period. “

Comment 17: Data analysis and result presentation: No clear or detailed plan for data synthesis and analysis is presented.

Response: We have now improved on our data analysis plan: 

“Median and interquartile range of all crude incidence rates will be reported. A Freeman–Tukey double-arcsine transformation will be applied to normalize data given an expected wide variation in extracted crude estimates from studies. Transformed data will be used to calculate summary proportion and 95% CI using a random effects model. A meta-analysis of the estimates of prevalence and incidence will be conducted if studies are sufficiently homogenous. We will assess heterogeneity using the X2 and I2 tests. I2 cut-off values of 0%, 25%, 50% and 75% will represent no, low, moderate and high level of heterogeneity [28]. We will conduct the meta-analysis following DerSimonian and Laird method [29]. Due to potential heterogeneity, DerSimonian-Laird random-effects model is preferred. DerSimonian and Laird method is a widely documented in the literature and is robust in various scenarios including skew-normal and extreme distributions for the effects [30]. However, where meta-analysis is not possible, a narrative synthesis of results will be done, using tables and figures as appropriate for data visualization. Also, in case of heterogeneity, we will carry out sub-group analysis and meta-regression analyses to show the potential sources of heterogeneity. Sub-group analysis will show the distribution across age group, sex and study regions. There should be at least 10 studies for each variable in the model for meta-regression analysis to be performed [31]. A sensitivity analysis will be performed to assess the robustness of the study estimates by assessing the effect of omitting a study at a time on the pooled estimate [31]. Lastly, a funnel plot asymmetry test and Egger’s test will be used to assess publication bias, if there are more than 10 included studies [31].”

Comment 18: Line 178: I suggest removing this sentence as it is already mentioned under "data items and description".

Response: The sentence has been removed.

Comment 19: Lines 181-183: The two sentences should be moved to the "study outcomes paragraph" as they represent a definition of the study's primary outcome (i.e., incidence).

Response: We have now moved the sentences to the study outcome paragraph as shown under comment 16.

Comment 20: Lines 182-183: How you are going to combine two incidence measures in the data synthesis and analysis while they have different denominators (i.e., either DM population at a country level or DM population presenting at healthcare facility). More clarification should be provided.

Response: Analysis will be conducted separately for population and hospital-based studies.

Comment 21: Lines 185-187: I think it would be good to add any potential reasons for not conducting a meta-analysis. A justification for using DerSimonian-Laird random-effects model should be stated. Also, will you consider any method for data transformation?

Response: A justification has been provided:

A meta-analysis of the estimates of prevalence and incidence will be conducted if studies are sufficiently homogenous. We will assess heterogeneity using the X2 and I2 tests. I2 cut-off values of 0%, 25%, 50% and 75% will represent no, low, moderate and high level of heterogeneity [28]. We will conduct the meta-analysis following DerSimonian and Laird method [29]. Due to potential high heterogeneity, DerSimonian-Laird random-effects model is preferred. DerSimonian and Laird method is widely documented in the literature and is robust in various scenarios including skew-normal and extreme distributions for the effects [30]”

Comment 22: Discussion: This section does not reflect the significance and novelty of the current study, as well as the implications of the future findings.

Response: We have revised this section as appropriate. The section now read:

“The burden of DM has been projected to grow exponentially in LMICs in the near future, while the quality of care remains poor [15]. The incidence of LLA among diabetics is considered one of the markers of diabetes care quality, while the prevalence helps us to understand the extent and depth of the disease burden. Also, the worldwide prevalence of LLA is varies. To the best of our knowledge, this proposed study will be the first to provide an estimate of the incidence and prevalence of LLA among persons living with DM in LMICs. An understanding of the LLA burden will inform targeted response. Also, its distribution across various socio-demographic groups, as defined by age or gender, for instance, may help in identifying populations at risk and designing appropriate interventions targeting those groups. Likewise, variance across countries may inform future research studies which might look at the DM care in settings with a lower incidence of LLA and identify lessons that may be translated to other resource-limited settings. This review will follow a systematic approach using a predefined protocol and following standard recommendations for data extraction [32]. Potential limitations that may be encountered include heterogeneity in the individual studies related to the quality of the study designs, definitions used, or methods of assessing outcomes. We will, however, provide a detailed narrative report highlighting these discrepancies and provide suggestions for improving future studies.”

Comment 23: Line 66: HbA1c was abbreviated without prior definition.

Response: We have now defined HbA1c

Comment 24: Line 59: Better to write it as "access to healthcare services"

Response: We have revised this section.

Comment 25: Lines 130-133: Be consistent in the letter case of keywords.

Response: Thank you for your observation. We have adjusted these. 

“A comprehensive systematic search of the literature will be conducted on five databases: MEDLINE, Embase, Scopus, CINAHL, and African Journals Online (AJOL) using a combination of MeSH terms and keywords such as “diabetes”, “amputation”, “lower limb amputation”, “lower extremity amputation”, “amputee”, “incidence “ and “population study”

---

## [Editor Report · Decision Letter 1]

30 Mar 2022

Lower limb amputations among individuals living with diabetes mellitus in low- and middle-income countries: a systematic review protocol

PONE-D-21-29205R1

Dear Dr. Ajayi,

Thank you for providing enough revision and details to satisfy the reviewers/editor comments. We’re pleased to inform you that your manuscript has been judged scientifically suitable for publication and will be formally accepted for publication once it meets all outstanding technical requirements.

Kind regards,

Daoud Al-Badriyeh

Academic Editor

PLOS ONE

---

## [Editor Report · Acceptance letter]

5 Apr 2022

PONE-D-21-29205R1 

Lower limb amputations among individuals living with diabetes mellitus in low- and middle-income countries: a systematic review protocol 

Dear Dr. Ajayi:

I'm pleased to inform you that your manuscript has been deemed suitable for publication in PLOS ONE. Congratulations! Your manuscript is now with our production department. 

Kind regards, 

on behalf of

Dr. Daoud Al-Badriyeh 

Academic Editor

PLOS ONE